# Developmental Dyslexia, Reading Acquisition, and Statistical Learning: A Sceptic’s Guide

**DOI:** 10.3390/brainsci11091143

**Published:** 2021-08-28

**Authors:** Xenia Schmalz, Barbara Treccani, Claudio Mulatti

**Affiliations:** 1Department of Child and Adolescent Psychiatry, Psychosomatics and Psychotherapy, University Hospital, Ludwig Maximilian University of Munich, 80336 Munich, Germany; 2Department of Psychology and Cognitive Sciences, University of Trento, 38068 Rovereto, Italy; barbara.treccani@unitn.it (B.T.); claudio.mulatti@unitn.it (C.M.)

**Keywords:** reading acquisition, individual differences, statistical power, simulation, causality

## Abstract

Many theories have been put forward that propose that developmental dyslexia is caused by low-level neural, cognitive, or perceptual deficits. For example, statistical learning is a cognitive mechanism that allows the learner to detect a probabilistic pattern in a stream of stimuli and to generalise the knowledge of this pattern to similar stimuli. The link between statistical learning and reading ability is indirect, with intermediate skills, such as knowledge of frequently co-occurring letters, likely being causally dependent on statistical learning skills and, in turn, causing individual variation in reading ability. We discuss theoretical issues regarding what a link between statistical learning and reading ability actually means and review the evidence for such a deficit. We then describe and simulate the “noisy chain hypothesis”, where each intermediary link between a proposed cause and the end-state of reading ability reduces the correlation coefficient between the low-level deficit and the end-state outcome of reading. We draw the following conclusions: (1) Empirically, there is evidence for a correlation between statistical learning ability and reading ability, but there is no evidence to suggest that this relationship is causal, (2) theoretically, focussing on a complete causal chain between a distal cause and developmental dyslexia, rather than the two endpoints of the distal cause and reading ability only, is necessary for understanding the underlying processes, (3) statistically, the indirect nature of the link between statistical learning and reading ability means that the magnitude of the correlation is diluted by other influencing variables, yielding most studies to date underpowered, and (4) practically, it is unclear what can be gained from invoking the concept of statistical learning in teaching children to read.

## 1. Introduction

In order to learn to read, a child needs to learn statistical regularities that are ingrained in their orthography: any orthography contains regularities, such as the same visual symbols mapping to the same sounds across words. It has been proposed that learning such orthographic regularities (as well as other types of statistical regularities) relies on a domain-general cognitive mechanism, called *statistical learning*. Statistical learning enables individuals to detect a probabilistic pattern from an input stream and to be able to apply this knowledge when they encounter a similar stimulus in the future [1].

As reading, almost by definition, requires sensitivity to regularities such as print-to-speech-sound correspondences, research has investigated the role of statistical learning as a predictor of reading ability. The idea is that a statistical learning deficit may be causally related to developmental dyslexia (hereafter: dyslexia; e.g., the work of [2]). While some researchers argue that statistical learning is causally related to reading [3,4], others remain sceptical [5,6,7,8].

The statistical learning deficit theory is by far not the only causal theory of dyslexia that is controversial: other studies have proposed lower-level perceptual, neural, or cognitive deficits as causes of dyslexia (e.g., auditory temporal sampling framework; pro: [9]; sceptical: [10]; visual magnocellular deficit hypothesis; pro: [11]; sceptical: [12]). Such proposed deficits are distal causes of dyslexia [13,14]. In the distinction between distal versus proximal deficits as causes of dyslexia, a proximal deficit refers to a deficit that is located in the cognitive reading system, and thus specific to reading: for example, if a child has poor knowledge of grapheme-phoneme correspondences, this is a proximal cause of the resulting poor reading ability. Distal deficits are causes of proximal deficits or causes of other distal deficits. In the case of statistical learning (as outlined in detail below), a statistical learning deficit (distal) could result in reduced ability to learn which letters can or cannot occur in one’s orthography (distal), which might prevent the build-up of an orthographic lexicon (proximal). Thus, in a metaphorical chain between a distal cause and dyslexia, each link acts as a distal cause except for the link(s) closest to the end-state of reading ability. As another example of a distal cause theory, the temporal sampling framework [9] proposes that a cause of dyslexia is a problem with neural synchronisation to speech sounds at the temporal frequency of syllables (distal). This leads to over-reliance on phonetic information (distal), which, in turn, reduces children’s ability to perceive phonemic categories, such as hearing the /p/ in “pin” ([p]) and “spin” ([p^h^]) as the same phoneme (distal). The failure to perceive phonemic categories affects the learning of grapheme-phoneme correspondences (proximal), as this requires children to understand that the phonemic categories map onto graphemes (in alphabetic orthographies).

We examine the statistical learning deficit theory as a case study of a distal deficit hypothesis. First, we critically evaluate the evidence for a statistical learning deficit as a cause of dyslexia. Then, we turn to the more general question of what issues need to be resolved that are common to distal cause theories. Overall, this will provide directions for future research aiming to provide a comprehensive theory of the causes of dyslexia.

## 2. Is Statistical Learning Related to Reading?

Before turning to the relationship between statistical learning and dyslexia, it is worth asking a related question: what is the role of statistical learning in the cognitive reading process? If statistical learning were to play no role in learning to read at all, we would not expect a causal relationship between statistical learning and dyslexia: any correlations would be purely epiphenomenal. Conversely, establishing a role of statistical learning in reading does not necessarily imply that statistical learning is always or even sometimes a cause of dyslexia: it may be that a minimal degree of sensitivity to statistical regularities, which all children possess, is sufficient to develop sound reading skills. Nevertheless, it is important to understand the relationship between reading and statistical learning, as it will unveil information about the causal pathway(s) that may link statistical learning to dyslexia.

At this stage, it is important to consider how we define statistical learning. Definitions tend to differ across authors (for discussions, see the work of [1,15]). Here, we adopt a broad definition and assume three features that define statistical learning. First, statistical learning occurs by routine exposure to input material. This is in contrast to rote learning, such as memorising the spelling of an unfamiliar word out of context. (We remain agnostic about whether statistical learning is strictly implicit in nature, such that it only occurs when there is no conscious effort on the side of the learner to extract regularities and/or awareness that such regularities exist.) A combination of both types of processes is likely to underlie learning in most real-life settings: for example, learning to read involves rote learning the letters and their pronunciations, and further learning more subtle regularities via exposure: for example, English-speaking children are generally taught that the letter *a* is pronounced /æ/ as in “cat”, but they learn via exposure that it is often pronounced /a:/ when it occurs at the end of a word, such as in the word “spa”. Second, when material is learned by statistical learning, the knowledge about it should reflect the statistics of the input. We focus on two types of statistics: frequency and consistency. In the context of reading, if *a* maps onto /æ/ across contexts, learning this correspondence will be easier when we encounter many instances of this pairing, and if *a* consistently maps onto /æ/, and not, say, 90% of the time onto /æ/ and 10% of the time onto /æɪ/. Third, statistical learning allows for generalisation: the ability to apply the learned knowledge to instances where it occurs in a different context. Thus, a child who has learned the pronunciations of words such as “wasp” and “swan” will read the pseudoword *wamp* as /wɔmp/ rather than /wæmp/, by generalising the context-sensitive grapheme-phoneme correspondence [*w*]*a* →/ɔ/to this new context. This makes statistical learning a very potent mechanism, especially for a highly complex process such as reading, as it allows the learner to break down a large amount of unfamiliar material into familiar chunks.

If we accept that reading material consists, at least partly, of statistical regularities and that a statistical learning mechanism is required to learn these regularities, it is true by definition that reading acquisition relies on statistical learning. We define reading as deriving phonology and meaning from an orthographic code. The relationship between orthography and phonology and semantics has been described as quasiregular [16,17]: In alphabetic orthographies, graphemes map onto phonemes in a systematic way (e.g., in English, *t* →/t/), though there are also instances where context needs to be taken into account to derive the correct pronunciation of a letter (e.g., c →/k/ as in “cat”, but c[i] →/s/ as in “circus”), or when the pronunciation is unpredictable altogether (e.g., the English word “Wednesday”, which has two silent letters). The relationship between orthography and phonology is thus statistical: readers pick up more subtle regularities by exposure [18,19,20,21] and are sensitive to how consistently a given letter cluster maps onto corresponding phonology [22]. The nature of the relationship between orthography and phonology varies across orthographies but is quasiregular in all orthographies studied to date. For example, in Chinese, the link between print and speech is less transparent than in alphabetic orthographies, but the pronunciation is often derivable from a phonological radical, which is the right or bottom part of many characters (e.g., the work of [23]).

The quasiregular relationship between orthography and semantics is less obvious. In alphabetic orthographies, the relationship between meaning and phonology (and hence the orthography, which has been designed to closely represent phonology) has been described as arbitrary. However, in inflectional and agglutinative languages, morphology provides a (quasi-)systematic link [17], and as long as a word consists of familiar morphemes, readers can infer its meaning even if they have never heard it before: for example, speech errors such as “misunderestimate” or “irregardless” are understandable to English speakers. Morphology and other letter clusters have been shown to provide information about word class to a reader, which also helps to narrow down the meaning of an unfamiliar word [24,25,26]. In addition, in Chinese, a non-inflectional language, orthography often explicitly represents semantics in the form of a semantic radical, which is often the left or top part of a character. Beyond morphology, a large-scale linguistic analysis has shown that there is a slight tendency for words that sound similar to have a similar meaning. In English, an example are words such as “glimmer”, “glisten”, and “glitter”. Such tendencies have been found in a wide range of languages [27,28]. Thus, all orthographies studied to date have been shown to have some degree of systematicity between print and meaning.

The relationship between meaning and print strengthens when we consider the sentence context, where upcoming words can be predicted with some accuracy (e.g., the work of [29]), and the meaning of an unfamiliar word can often be guessed from the context. Overall, however, orthography-semantics statistical patterns are mostly not strong enough for readers to be able to infer the meaning of an unfamiliar word, as any adult learner of a foreign language can attest. Nevertheless, some degree of generalisation can occur: for example, most readers can not only pronounce but also make some sense of Lewis Carroll’s poem “Jabberwocky”, which consists almost entirely of pseudowords: “‘Twas brillig, and the slithy toves/Did gyre and gimble in the wabe [...].”

Thus, it seems that learning the relationship between orthography, phonology, and semantics relies on statistical regularities, and learning these should be indispensable for the reading process. In addition, reading relies on secondary skills, which are also heavily reliant on statistical learning. Children’s reading skills are predicted by oral language skills, such as phonological awareness [30] and vocabulary size [31], to name a few examples. The ability to use transitional probabilities between syllables has been linked to spoken word learning: even infants are able to discriminate between syllables that follow each other often, signifying that they are likely to be part of the same word, versus syllables that rarely co-occur, signifying that they are likely to be separate words [32]. This has been proposed to help vocabulary learning [33]. Statistical learning ability has also been linked to phonological awareness [34], and it has been considered in relation to grammar skills [35].

As another secondary skill, visual processing may also be a link between statistical learning ability and reading. Orthographic code consists of sequences of graphemes, and the sequences themselves contain statistical information: some graphemes are more likely to follow one another (e.g., in English, the letter bigram *am*, which occurs in 723 words; [36]), and other letters never occur next to each other (e.g., in English, *xm*). Other statistical patterns can also be used to make a word look nonword-like: for example, some letters are more likely to occur in certain positions of words: English words never begin with *ck* as a representation of the sound /k/, though this letter cluster often occurs in the middle or end of words. Sensitivity to such graphotactic regularities might help with spelling: for example, a child who is unsure how to spell the word “quick” can rely on her knowledge of these regularities to decide that it is unlikely to be spelled “ckwikk” [37]. There are several ways in which sensitivity to graphotactic regularities may aid the reading process, for example, by allowing readers to use their statistical knowledge to predict the identity of an upcoming letter that has not yet been processed (summarised in the work of [38], but see also the work of [39]).

In summary, to the question of whether statistical learning, broadly speaking, plays a role in reading, the answer is a clear “yes”. Given the preceding paragraphs, however, a sceptical reader will be wondering if this is a meaningful question. The frequency and consistency of input statistics affect learnability across all domains. Thus, much of the material that humans (or, in fact, any animals) learn is learned statistically by our definition above. Although statistical learning seems to occur across domains, it is unclear whether, in practice, the process of statistical learning can be dissociated from the modality in which it is observed [40,41]. Thus, an observed deficit in the statistical learning of reading-related material may reflect a general processing deficit in this domain, rather than in the cognitive structures responsible for extracting regularities (we thank an anonymous reviewer for pointing this out).

To make the question, if statistical learning plays a role in reading, meaningful in the context of dyslexia research, there are two ways to further specify it: (1) Is there a domain-general statistical learning mechanism that affects, among others, reading acquisition and which may cause dyslexia if impaired? (2a) Are there any cognitive components of the reading process that are particularly reliant on statistics, and (2b) do children with dyslexia have a problem in learning these statistics, which, in turn, affects their ability to read?

For the first question, it seems a priori unlikely (to a sceptic) that the answer should be “yes”. After all, if statistical learning is involved in most (or all) learning tasks, we would expect that children with dyslexia should also be impaired in domains ranging from language [32] to social interaction [42] (see the work of [43] for a discussion of this issue). While many correlates of dyslexia have been documented across domains (e.g., motor procedural learning; [44]), it is possible, for example, for language disorders to occur in the absence of any reading problems and vice versa [45]. It is not even clear that statistical learning can be described as a single process: recent studies support a componential view, where different types of statistical regularities and in different domains depend on different processes [40,41].

In terms of providing evidence that would directly address the question, one would need to find a correlation between reading ability and statistical learning in a non-reading-related domain or a group difference in statistical learning ability between individuals with dyslexia and a control group. While such studies exist, the results are mixed (e.g., positive: [4]; negative: [6]), and systematic reviews and meta-analyses have concluded that there is insufficient evidence for such a link [5,7]. Methodological issues include publication bias and low reliability when it comes to most statistical learning tasks [15,46]. Due to these theoretical and empirical issues, therefore, we consider it unlikely that studying this question would yield fruitful results and focus, for the remainder of the article, on the second question.

To address the second question about the specific components of reading that rely on learning statistics, we do not need to rely on the controversial assumption that there is a single domain-general statistical learning process. Instead, we can focus on different aspects of the reading process that contain regularities and examine whether (1) individual differences in the degree of sensitivity to these specific statistical regularities predict reading ability and (2) whether individuals with dyslexia show reduced sensitivity to these regularities.

## 3. Making the Links in a Causal Chain

As discussed in the previous section, many aspects of reading rely on statistical regularities. In order to identify a potential statistical learning deficit that causes dyslexia, we need to consider causal pathways that would link a statistical learning process to the behavioural outcome of poor reading ability. Between the cognitive statistical learning process and the behavioural outcome of reading ability or dyslexia, there needs to be at least one intermediate step in the form of knowledge, which is acquired via statistical learning and that affects the reading acquisition process, either directly or via additional links in a causal chain. In order to establish a causal link with dyslexia, one needs to show that the absence of this knowledge impedes reading acquisition.

### 3.1. Graphotactic Knowledge as an Intermediary Link

Previous studies have focussed on empirically establishing parts of such causal chains. One possible causal chain involves graphotactic knowledge as an intermediary: the knowledge about letter clusters and the positions in which they can occur. It has been known for decades that readers across different European orthographies are sensitive to such statistics (e.g., the work of [47,48,49]). This knowledge must have been acquired via statistical learning, as per our definition: First, reading instructions generally do not include explicit instructions about common letter clusters and positional regularities; therefore, this knowledge, in most cases, must have been learned via exposure. Second, the knowledge is statistical, as it reflects the frequency and consistency with which certain letter clusters co-occur. Third, readers generalise their knowledge of graphotactic regularities, as they can determine whether nonwords that they have never seen before contain illegal letter clusters.

While the link between statistical learning and graphotactic knowledge seems clear, the next question is whether graphotactic knowledge is causally related to reading ability and whether a lack of graphotactic knowledge leads to dyslexia. There are alternative explanations for a correlation between graphotactic knowledge and reading ability: First, it is possible that graphotactic knowledge is epiphenomenal to reading: by being exposed to printed words, readers automatically extract knowledge about legal letter combinations and positions, but this knowledge may play no causal role in how well they read. The direction of causality would instead be from reading ability to graphotactic sensitivity: the better readers with more exposure to print may be more sensitive to regularities due to an increased input [50]. Second, it is possible that there is insufficient variability in children’s graphotactic knowledge: a minimum level of knowledge might be sufficient to facilitate reading acquisition, and this level may be so low that all children are able to reach it within a short period of time. Empirically, a correlation between graphotactic sensitivity and reading ability in beginning readers is well established [47,48,49,50]; in adults, the reading process seems to be no longer facilitated by the presence of high-frequency clusters [39]. However, the correlational design does not allow us to draw causal conclusions: a correlation may indicate that graphotactic sensitivity is epiphenomenal.

Several studies have circumvented this issue with correlational designs by using a learning paradigm with artificial orthographies. In artificial orthography learning studies, participants are exposed to symbol strings written in unfamiliar symbols. This has two advantages: first, it allows for exact control over the statistical properties of the input material, and second, the researcher can track the learning process as it occurs. In such studies, the experimenter can vary the frequency with which certain letter clusters occur in the training material. Participants are sensitive to this manipulation: in a subsequent test phase, they are more likely to categorise a novel symbol string as familiar when it contains a high-frequency cluster [51,52]. Furthermore, when participants also learned the pronunciation of pseudowords, their degree of sensitivity to graphotactic regularities correlated with their accuracy in subsequent reading aloud tasks of those pseudowords [51]. This suggests that knowledge of graphotactic regularities may facilitate the learning of print-to-speech correspondences. However, as the frequency of the letter clusters correlated with the frequency with which participants heard the letter-speech pairings during the training, further studies are needed to establish how purely visual knowledge of graphotactic regularities affects reading acquisition.

Establishing the role of graphotactic regularities in unimpaired reading is an important first step in understanding the relationship between graphotactic sensitivity and dyslexia. Taken together, the correlational and artificial orthography studies show that readers are sensitive to graphotactic regularities and that such sensitivity is acquired easily after exposure to a new script. Despite the experimental approach of the artificial orthography learning studies, however, it is still unclear whether graphotactic sensitivity plays a causal role in facilitating reading acquisition.

An additional open question is to what extent these studies inform us about the relationship between dyslexia and statistical learning, as their focus is on reading ability in unselected samples of adults and children. In studying graphotactic sensitivity in adults and children with dyslexia, researchers have turned to the artificial grammar learning task [53]. Here, participants are exposed to letter or symbol strings. These strings are created by a set of rules about which symbol(s) can follow another. These rules thus create graphotactic regularities. In a training phase, participants have been shown to learn these graphotactic regularities: in a subsequent test phase, they can distinguish grammatical from non-grammatical sequences, even when they did not occur in the training materials, showing generalisation. If dyslexia is caused by a reduced ability to pick up graphotactic regularities, participants with dyslexia should show lower learning performance than a control group on this task. The classical artificial grammar learning task used letter strings as stimuli: for dyslexia research, this creates a confound between letter processing and the ability to pick up the regularities [54]. Studies on participants with dyslexia, therefore, use non-letter symbols and show mixed results when it comes to a group difference between participants with and without dyslexia (e.g., positive result: [55]; negative result: [56]). The literature seems to be affected by publication bias and questionable research practices [5,7]. Thus, while a meta-analysis has shown an overall group difference, further evidence is required to convince a sceptical researcher.

A recent study has assessed the role of graphotactic knowledge for spelling in a more ecologically valid setting [37]. Spelling ability has previously been linked to orthographic learning: orthographic learning refers to the knowledge of a written word form, which is important for fluent reading [57,58]. Zhang and Treiman [37] asked preschool children with no knowledge of letters’ phonology to perform a delayed copying task: the children were visually presented with nonwords which either contained common letter bigrams (e.g., *CHED*) or illegal bigrams (e.g., *EHDC*). After the nonword disappeared, the child was asked to spell the word they had just seen. Indeed, children provided more accurate spellings for nonwords with legal letter bigrams, suggesting that children are sensitive to graphotactic regularities. Due to the use of the spelling task, this experiment provides evidence that is in line with a causal chain from sensitivity to graphotactic regularities to spelling and reading ability. However, a limitation of the study is that the participants were pre-readers. As such, it is not clear whether children who are already undergoing reading instructions continue to rely on graphotactic knowledge for spelling. Once children learn about the relationship between graphemes and phonemes, this new knowledge is likely to become a stronger predictor of the likely spelling of a word. This may reduce the extent to which they rely on graphotactic regularities. Thus, it remains an open question how graphotactic regularities affect reading acquisition in children after they reach the alphabetic phase of the learning process [59].

In summary, the first proposed causal pathway could lead from statistical learning to graphotactic knowledge to spelling ability; spelling ability should lead to more efficient orthographic learning, which, in turn, is important for fluent reading [57,58]. The existing studies provide important building blocks: they unequivocally show that children and adults become aware of graphotactic patterns already after a small amount of exposure. They also provide evidence that this knowledge may be involved in reading and spelling acquisition, especially at the beginning of the learning process. However, it is still unclear whether variability in the degree of graphotactic sensitivity is related to reading acquisition after children reach an alphabetic phase. Furthermore, and of critical importance to the topic of the current article, there is, to date, no reliable evidence that would suggest that low sensitivity to graphotactic regularities may lead to dyslexia.

### 3.2. Learning Orthography-Phonology Mappings

In the literature on dyslexia, the pseudoword reading aloud deficit is one of the few consistent and uncontroversial findings: children with dyslexia perform consistently worse than control groups at pseudoword reading aloud tasks [30]. Pseudoword reading ability can be seen as a measure of the extent to which readers can generalise their knowledge of grapheme-phoneme correspondences. In English, the basic correspondences (e.g., *t* is pronounced /t/) are generally explicitly taught; the knowledge of such basic correspondences is, therefore, a candidate of a type of knowledge that is not learned statistically, as the exposure is deliberate rather than incidental. However, more subtle regularities are picked up via incidental exposure [19,21]. In orthographies such as English, German, and Dutch, the pronunciation of a given grapheme sometimes depends on the context [60]. This can be described by context-sensitive correspondences: for example, in English, *c* →/k/, but *c*[*i*] →/s/, or *a* →/æ/, but [*w*]*a* →/ɔ/. Pseudoword reading aloud experiments have consistently shown that adult readers often rely on such correspondences, for example, by pronouncing the pseudoword *wamp* as /wɔmp/ rather than /wæmp/ (e.g., the work of [18,20]. Thus, generalisation can and does occur. A study in Dutch has shown that, when asked, adult readers were unable to explain why they preferred a context-sensitive pronunciation (/s/) for pseudowords starting with *c*[*e*,*i*], as opposed to the default pronunciation for *c*,/k/[61]. This further strengthens the notion that such regularities are learned implicitly and without explicit instruction. Thus, context-sensitive correspondences seem to be learned via a statistical learning process.

We remain agnostic about whether knowledge about context-sensitive grapheme-phoneme correspondences is stored in the cognitive system in the form of all-or-none rules, as is implemented in the dual-route cascaded model of reading aloud [62], or if it is graded and represents the statistical properties of letter clusters and their corresponding phonology from the input material, as in connectionist models [16,63]: in either framework, extracting knowledge about grapheme-phoneme correspondences across contexts can be seen as a statistical learning process. The connectionist perspective can be reconciled more easily with a statistical learning perspective, as the format of knowledge should reflect the statistics of the input material, even in skilled adult readers. In a rule-based framework, the statistics should play a role during the learning process, as a regularity that is encountered often is more likely to be stored as a rule. However, after a rule is established, the frequency or consistency with which this rule is complied with in the reading material becomes irrelevant. Sensitivity to statistics in the input materials, in the form of consistency effects, is well-documented (e.g., the work of [22]). The consistency effect describes the finding that when a word or pseudoword contains an orthographic unit that has several plausible pronunciations (e.g., *-int* in *mint* versus *pint*), it takes longer to process than a word where all sublexical units have a consistent pronunciation. The size of the consistency effect has been shown to be correlated with reading ability in children [64].

In many orthographies, orthography-phonology regularities also exist on a supra-segmental level. Lexical stress assignment has been studied in orthographies, ranging from English (e.g., the work of [26]) to Russian [65] and Italian [66]. Readers have been shown to rely on hierarchically organised regularities abstracted at multiple levels. On a basic level, they show a tendential preference towards the most common stress pattern in their language when reading aloud polysyllabic pseudowords. The most common stress pattern, in many orthographies, varies as a function of word class; readers also pick up on this regularity and, when the grammatical class of a pseudoword is indicated by its morphology, prefer a pronunciation that is in line with the word class. Beyond morphology, readers have a preference for stress assignment, which is the same as orthographically similar words. This shows a level of regularity extraction and generalisation relying on complex, subtle regularities.

Regularities underlying segmental orthography-phonology knowledge and stress assignment do not exist in all orthographies: for example, stress assignment is fixed in orthographies such as French, and, by definition, non-alphabetic orthographies have a qualitatively different way of conveying the pronunciation of a word from its spelling. However, all orthographies convey some regularities. For example, in Chinese, orthographic regularities exist in constraining the position of radicals within a character, where semantic radicals tend to occur in the left or top position of a character, and phonetic radicals tend to occur in the right or bottom position of a character. Children develop a sensitivity to these regularities [67]. The current article does not provide an exhaustive list of all orthographic and linguistic regularities that exist across languages; however, it is safe to assume that the presence of such regularities is a global phenomenon.

It is intuitive that knowledge of context-sensitive correspondences and other orthography-phonology regularities should, in turn, affect pseudoword reading aloud. A correlation between the application of context-sensitive correspondences and overall reading ability has been established [21]. As such, a possible causal chain could go from impaired ability to learn the regularities between orthography and phonology to poor decoding ability. Decoding is necessary for reading any word that is unfamiliar in its written form. As most words are visually unfamiliar to a beginning reader, an impaired ability to decode should hamper reading acquisition [68]. The evidence for a link between knowledge and application of context-sensitive rules and reading ability, however, is correlational: thus, again, we cannot conclude that reduced sensitivity to these regularities causes poor reading ability, or if poor reading ability is associated with reduced exposure to text, and thus leads to reduced opportunity for readers to extract such regularities. The involvement of sensitivity to orthography-phonology regularities in dyslexia remains an open question: as this sensitivity seems to covary with reading ability across a wide range of reading skills, it is unlikely that it has a unique role in dyslexia.

### 3.3. Learning Orthography-Semantics Mappings

As discussed above, there are some subtle regularities between orthography and semantics. Reliance on morphology has already been studied to a great extent, both in relation to reading processes in general and to dyslexia. Knowledge of morphology is not specific to reading: it provides a source of regularity both in the written (orthography-semantics) and spoken (phonology-semantics) form: a speaker of English is likely to understand pseudowords made of familiar morphemes, such as “misunderestimate”, regardless of whether they hear someone use it orally or whether they see it written. In alphabetic orthographies, the relationship between orthography and phonology is sufficiently close that we can consider the orthography-to-semantics and phonology-to-semantics information to be mostly overlapping. It is important to bear in mind, however, that children are exposed to morphological regularities before they start learning to read.

Across orthographies, morphological awareness seems to be related to reading ability, with children who have higher scores on morphological awareness tasks showing better reading ability (e.g., the work of [69,70]). When it comes to the literature on dyslexia, in alphabetic orthographies, several studies found intact or even superior morphological skills in adult samples with dyslexia, compared to control groups [71,72]. This suggests that reliance on orthography-semantics regularities may develop as a compensatory mechanism for poor reading ability.

Beyond morphological awareness, further studies provide some evidence for knowledge of orthography-semantics mappings acting as a compensatory mechanism [64,73,74]. These studies used the imageability effect to assess reliance on orthography-semantics mappings. The imageability effect refers to the finding that words for which the meaning is easy to visualise (e.g., “piano”) are processed faster for words with more abstract meanings (e.g., “music”). These studies found an inverse relationship between reading ability and the size of the imageability effect, with poorer readers showing a bigger effect. As the imageability effect does not measure knowledge of systematicity between orthography and semantics, this finding alone cannot be taken as evidence for intact learning of regularities on the orthography-semantic level.

The question of whether readers rely on semantic regularities beyond morphology is only beginning to be answered (e.g., the work of [75,76]). This is partly due to the development of novel ways to capture the relationship between orthography and semantics [77]. Thus, it is not yet clear how the sensitivity to learning orthography-semantic regularities is related to reading ability or dyslexia.

Orthography-semantics knowledge can be acquired and stored in two ways: Any arbitrary associations need to be learned by rote and thus do not represent a statistical learning process. However, systematic relationships, such as those underlying morphologically complex words, can be learned and stored via their underlying regularities. It is an open empirical question in which experimental paradigms can be used to dissociate these two processes. Artificial language learning studies may be able to manipulate whether the meaning of a pseudoword can be derived from the knowledge of similar pseudowords. Further empirical investigations can assess whether psycholinguistic marker effects can be dissociated. Candidates for psycholinguistic marker effects reflecting statistical orthography-semantics knowledge include the recently developed orthography-semantics consistency measure: the extent to which words that have similar spellings have similar meanings [77].

In relation to dyslexia, a first hypothesis is that orthography-semantics regularities are not learned due to an underlying statistical learning deficit, and readers learn to rely on their orthography-semantic rote knowledge to compensate both for their impaired knowledge of orthography-semantic regularities and of orthography-phonology regularities. A second hypothesis is that the causes of dyslexia are related specifically to the learning of orthography-related material. Children are exposed to phonology-semantic regularities even before reading acquisition starts: if the learning deficit associated with dyslexia is limited to orthographic material, then children learning to read in alphabetic orthographies should be able to directly transfer their knowledge of the phonology-semantics regularities to relying on orthography-semantics regularities, provided they have reached a base level of decoding ability.

The first hypothesis would predict that participants with dyslexia should be impaired at learning orthography-semantics regularities but may instead show stronger rote knowledge, which they develop as a compensatory mechanism. According to the second hypothesis, there should be a dissociation between knowledge of orthography-semantic regularities and rote knowledge. As this distinction has not been explicitly and empirically studied to date, we do not have the data to support one hypothesis over the other. From the morphology literature, it is unclear how to reconcile the finding of a positive relationship between morphological awareness and reading ability in unselected samples of children, but no group difference in morphological awareness between adults with and without dyslexia. Future empirical studies should seek to resolve this inconsistency.

### 3.4. Interim Summary

We can conclude that, beyond doubt, (1) orthographies contain regularities, (2) readers learn these regularities via a mechanism that, by our definition, qualifies as “statistical learning”, and (3) their sensitivity to these regularities largely correlates with their reading ability. However, we remain sceptical about the claims (1) that the relationship between readers’ sensitivity to these regularities and their reading ability is causal, and (2) that a statistical learning deficit is a cause of dyslexia in some or all cases.

## 4. Issues in Establishing Distal Causes: A Noisy Chain Hypothesis

The above section describes two possible causal chains between statistical learning and reading ability or dyslexia. In both of these, statistical learning would act as a distal cause of reading ability and a distal deficit associated with dyslexia. As noted in the Introduction, other distal cause theories have been put forward, such as a procedural motor learning deficit [44], an auditory temporal sampling deficit [9], and a visual magnocellular deficit [11]. All of these theories remain controversial, and the evidence is mixed. In contrast, theories that focus on proximal causes are less controversial: for example, it is well established that children with dyslexia have problems with phonological processing (e.g., the work of [30,78]; though note that even here, it is controversial whether a phonological deficit is causal to dyslexia; [79]). Furthermore, in treatment studies of children with dyslexia, training children on cognitive mechanisms that have a proximal connection with reading ability is generally more effective than treating distal deficits [14,80].

While the above section has focussed on methodological and empirical issues specific to statistical learning, here we describe a theoretical-statistical issue that is relevant to all distal cause theories. The acknowledgement of this issue can help shed light on why the distal cause theories tend to be controversial. In a metaphorical causal chain between a perceptual (e.g., auditory), neural (e.g., neural synchronisation to speech), or low-level cognitive (e.g., statistical learning) process and reading, we expect the connection between each link, representing a cognitive skill, will not be deterministic, in the sense that a deficit in a link will not necessarily lead to dyslexia. For example, a child’s statistical learning ability is unlikely to be the only determinant of their graphotactic sensitivity; one obvious additional determinant is the amount of exposure to printed materials. This additional determinant dilutes the correlation between statistical learning ability and graphotactic sensitivity: even if we develop measures of both constructs that are completely free of measurement error, we will not obtain a correlation of 1 between them. The same holds true of the relationship between sensitivity to graphotactic regularities and spelling ability: even if we assume that this relationship is causal, other factors, such as the child’s phonological decoding ability, will additionally contribute to determining their spelling ability. Thus, as we travel further down the chain, the relationship between our first distal cause (e.g., statistical learning or the perceptual impairment following a magnocellular deficit) and reading ability or dyslexia will be diluted, which will be manifested as a reduced correlation between increasingly distal skills.

Theoretically, it should be possible to identify all contributors of a single skill or link, but we are still far from such a comprehensive understanding of the predictors of reading ability and dyslexia. If we treat all unknown contributors of a skill as random noise, it becomes possible to model the correlation between a distal skill and the end-state outcome of reading under a range of assumptions. We expect that such modelling will show two dependencies: (1) The greater the number of links between a distal cause and the end-state of reading, the smaller the correlation between the most distal cause and reading ability, and (2) the smaller the relative contribution of Link *L*-1 to Link *L*, the smaller the correlation between the most distal cause and reading ability.

The extent to which Link *L* contributes to Link *L*+1, relative to the additional predictors, is likely to vary. Some links may be very tight, such as knowledge of grapheme-phoneme correspondences as a predictor of pseudoword reading ability: though other skills (such as the ability to blend phonemes after successful conversion) are likely to affect pseudoword reading ability, this error term will be relatively small. For other links, the causal relationship may be relatively weak: if we assume that knowledge of grapheme-phoneme correspondences is affected by statistical learning ability, it will also be affected, for example, by the amount of exposure, explicit instruction of basic or even more complex grapheme-phoneme correspondences. We can model this by multiplying the contribution of Link *L*-1 and the noise term by *x_L_* and *y_L_,* where *x_L_* + *y_L_* = 1. In the case of the link between knowledge of grapheme-phoneme correspondences and reading ability, the contribution of grapheme-phoneme correspondence knowledge (*x_L_*) might be 0.8, with the remaining 20% (*y_L_* = 0.2) reflecting the influence of all other predictors. In the case of statistical learning to the knowledge of grapheme-phoneme correspondences, this contribution will be much smaller (e.g., *x_L_* = 0.2), with the remaining 80% reflecting the influence of all other predictors (*y_L_* = 0.8)

We will model each Skill *X* as a standard normal distribution (Skill *X* ~ *N*(0,1)). Skill *A*, denoting the most distal cause, is thus modelled by drawing numbers from a standard normal distribution (μ = 0, σ = 1). Skill *B* is partly determined by Skill *A* and partly by noise (ε), resulting from additional, unknown predictors, again, modelled as a standard normal distribution (ε*_n_* ~ *N*(0, 1)). Taken together, then, the model, for *n* amount of links, including the most distal cause (Skill *A*), will look as follows:Skill *A* ~ *N*(0, 1)
Skill *B* = x_1_ ∗ Skill *A* + y_1_ ∗ ε_1_, ε_1_ ~ *N*(0, 1)
Skill *C* = x_2_ ∗ Skill *B* + y_2_ ∗ ε_2_, ε_2_ ~ *N*(0, 1)
…
Reading ability = x*_n_* ∗ Skill *n* + y*_n_* ∗ ε*_n_*, ε*_n_* ~ *N*(0, 1)

We implemented this model in R (version 4.0.2; [81]); the code can be found on the Open Science Framework: https://osf.io/epu8m/ (accessed on 1 June 2021). We varied both the number of links (*n*) and the relative contributions of Link *L* versus the unknown predictors (*x* and *y*). The number of links varied between a direction relationship (Skill *A* → reading) with no intermediate links, and 1 (Skill *A* → Skill *B* → reading), 2, 3, or 4 intermediate links.

For values of *x*, we arbitrarily chose different values for scenarios where the contribution of Link L to Link L+1 (*x_n_*) may be low (0.2), medium (0.5), or high (0.8). This value is likely to vary for each link within a chain; however, to keep the number of simulations with different parameters manageable, we kept it constant within each chain (i.e., for all links within a chain, *x_n_* is either 0.2, 0.5, or 0.8). For each scenario, we simulated 1000 participants’ scores on Skills *A* to Skill *n*, as well as reading ability, and calculated the correlation between Skill *A* (the most distal cause) and reading ability. We standardised each skill to retain the interpretability of the scores as *z*-scores. To derive a confidence interval in addition to a point estimate for the correlation, we repeated this process, for each scenario, 10,000 times (the number of simulated participants and simulations were chosen within a trade-off between getting reliable output values and computational restraints). This allowed us to provide a point estimate for the correlation (average correlation across 10,000 simulations) and an upper and lower bound of a 95% CI (bounds of the part distribution that contain 95% of the simulated data). These are depicted in Table 1. From the correlation coefficients, it is trivial to provide a power calculation: how many participants are required to detect the correlation with *p* < 0.05, 80% of the time, for the given correlation? These calculations were conducted with the R package pwr (version 1.3-0; [82]).

We note that the scenarios with a strong relationship between Link *L* and Link *L*+1 are likely to be overly optimistic: in most real-life cases, additional skills would account for more than 20% of performance on Skill L+1. Furthermore, the current simulations ignore the role of measurement error for the sake of simplicity: measurement error further reduces the correlations between any set of skills. The causal chains for statistical learning that we have described above contain, as a minimum, two intermediate links: for example, statistical learning (Skill A) → sensitivity to complex grapheme-phoneme correspondences (Skill B) → ability to read unfamiliar words (Skill C) → reading ability. In this case, assuming a medium-strength relationship (which seems to us like the most optimistic but still realistic case scenario), one would need 170 participants to detect a correlation between statistical learning and reading ability. To name some recent examples, correlational studies that have assessed this relationship tested 38 children and 37 adults [4], 84 adults [6], 65 children [83], 72 adults and children [84].

In dyslexia research, rather than calculating correlations, researchers select a sample of dyslexia and compare their performance on Skill *A* to a matched control group. We simulated this scenario to derive an effect size estimate: instead of calculating a correlation between Skill *A* and reading, we divided the simulated participants into dyslexic participants (standardised reading score *z* < −2) and controls (standardised reading score *z* ≥ −2) and derived a standardised effect size (Cohen’s *d*) by subtracting the mean simulated performance on Skill *A* of the control group from the mean simulated performance on Skill *A* of the group with dyslexia, and dividing the difference score by the overall simulated population standard deviation (*d* = (Skill *A* mean _dyslexia_ − Skill *A* mean _control_)/Skill *A* standard deviation _all_). To increase the number of participants in the dyslexia group for reliable statistics, we changed the number of simulated participants to 10,000. Otherwise, the simulation and parameters were identical to the previous simulations. The results of these simulations are presented in Table 2.

An optimistic-realistic-case scenario (medium-strength relationship, 2 intermediate links) shows that, in order to detect a significant group difference between participants with and without dyslexia in the most distal deficit, we would need 45 participants (approximately 23 participants per group). Again, most studies have a smaller sample size: in a systematic review of artificial grammar learning studies in participants with and without dyslexia, we have found a median sample size of 32 participants (16 per group), with a minimum of 30 and a maximum of 223 in both groups [5].

## 5. Directions for Future Research

So far, we have identified both theoretical and statistical issues pertaining to the statistical learning theory of dyslexia. Some of these can be applied to other distal theories of dyslexia. From a statistical perspective, the noisy chain hypothesis and our simulations thereof clearly show that most studies, to date, are underpowered to detect a correlation between a distal cause and the end-state outcome of reading ability, or a group difference in measures of the distal cause between participants with and without dyslexia. This is an issue that is common to all distal theories of dyslexia. Across simulations, we showed a wide range of correlations and effect sizes, resulting in vast differences in sample size calculations for achieving adequate power. Future research could aim to narrow these down by establishing plausible parameter values. These are likely to depend on the particular research question.

For a researcher proposing a distal theory, the implication of this finding is that the number of links in the causal chain should be taken into account when determining sample size. This will improve the research on the causes of dyslexia in two ways. First, this will force the researcher to specify an exact causal pathway. Second, it is likely that the sample size required to find a link between the most distal cause and reading ability will be much larger than what is generally used in this kind of research. The specification of the causal pathway will have a theoretical benefit, as it will avoid ambiguity, which results in unclear predictions, thus making it difficult to either support or falsify a given theory.

Improving sample size, overall, will improve our ability to draw conclusions from mixed sets of findings. Low statistical power is problematic in most areas of psychology, with no apparent improvement, on average, since the phenomenon was first described in the 1960s [85,86]. A commonality between distal causal theories of dyslexia is the mixed empirical results, with failures to replicate an initial, often positive finding of a group deficit. By definition, we would expect that many studies would yield non-significant results when we have low statistical power. Thus, the false-negative rate is likely to be high. However, other common issues in social and behavioural sciences increase the false positive rate: for example, publication bias [87] and questionable research practices [88]. This leaves us in a situation where it is very difficult to interpret a mixed set of findings, as both a true effect with false negatives due to low power and the absence of a true effect with some false positives are plausible scenarios.

An obvious solution to the problem of low statistical power is to run larger studies. Depending on the scenario, however, this may require sample sizes in the thousands or even up to 200,000,000 participants. Due to the practical hurdles of testing such a big number of participants, this might not always be feasible. To circumvent this issue, researchers may take a more experimental approach when possible. In artificial orthography learning studies, we can minimise the effect of existing print-related knowledge, which, in itself, is likely to be caused by a plethora of known and unknown, measured, and unmeasurable external variables. While such experiments lack external validity, they can lay the groundwork for a follow-up study [89]: if a researcher can show a model of the causes of dyslexia and support the links of their causal chain with experimental evidence, they can use this knowledge to apply for an amount of funding that will allow them to perform an adequately powered study.

Another recommendation is to focus on individual links in the causal chain instead of focussing, from the very beginning of a new line of research, on the relationship between the most distal skill and reading ability. The correlation between two adjacent links is likely to be strongest. From a theoretical perspective, preliminary studies focussing on only a part of a causal chain can be used to provide evidence for or against the proposed theory before a large-scale observational study is conducted. The findings of the study can also be used to estimate the parameters in the simulation above: the strength of the relationship between individual links and the number of linking skills. This will allow for a more accurate effect size estimate. Furthermore, the methods of preliminary studies can be tweaked to minimise measurement error, leading to stronger correlations and a smaller number of participants, which will be required for a large-scale observational study.

### 5.1. Establishing Causality for the Statistical Learning Deficit Theory

In the section on making links in a causal chain for the statistical learning deficit theories of dyslexia, we have shown correlational evidence but concluded that evidence for causality is missing. Again, this is an issue that is relevant not only for the statistical learning deficit theory but also for other distal theories, which are often correlational.

In the case of statistical learning and dyslexia, previous studies have already provided important puzzle pieces towards identifying the direction of causality between sensitivity to orthography-related regularities and reading ability. Experimental studies with artificial orthographies are able, in principle, to establish causality, but they are limited in their ecological validity: it remains unclear to what extent adults learning an artificial script use similar cognitive mechanisms as children learning to read. Correlational studies can be conducted with children learning to read in a natural setting but cannot be used to establish causality. The missing link to establish causality would be either a longitudinal study or a training study. Such studies would need to be carefully designed to truly understand the direction of causality: children who are inherently good at reading tend to read more [90], which means that better readers have more exposure to texts and the regularities that they contain [91].

A longitudinal study would be able to shed further light on the relationship between statistical learning, knowledge of print-related regularities, and reading. Such a study should start with preschool children, akin to Zhang and Treiman [37]. The first question is whether individual differences in the ability to rely on graphotactic regularities in spelling continue to predict reading acquisition after the onset of formal instruction. However, even here, we cannot exclude the reverse direction of causality: even if children have not started reading acquisition yet, their knowledge of graphotactic regularities must come from having been exposed to print. Such a study would therefore not provide an unequivocal link between statistical learning ability and reading but would at best show that knowledge of statistical regularities aids reading acquisition. A second question is whether preschool children’s ability to extract regularities from an artificial orthography would predict their later reading abilities in their native language. If a statistical learning deficit causes dyslexia, one would predict that all children who are impaired in extracting regularities will develop dyslexia after the onset of reading instructions. This combination of experimental and observational research would help to establish a causal link between the ability to learn regularities and reading acquisition.

What would be the implications of further research on statistical learning and dyslexia, and what are the implications of the findings so far? The studies to date have provided important theoretical insights into learning mechanisms that potentially underlie not only reading but also other visual learning tasks [92]. Understanding the direction of causality would shed further light on the cognitive mechanisms underlying reading acquisition. This would help tighten the link between linguistics and educational psychology, as it would open directions for linguistic research on relevant regularities that are present in orthographies and spoken languages.

### 5.2. Practical Implications of the Statistical Learning Deficit Hypothesis

The practical implications of statistical learning and dyslexia research are likely to be more limited. In general, identifying a causal deficit is associated with the promise of early identification and treatment. To date, we lack any indication that general statistical learning ability is causally related to reading ability. If such evidence is found in future studies, it will be a challenge to devise statistical learning tasks for children with psychometric properties that would allow for them to be used as a diagnostic test [46]. The link that does have empirical support is between sensitivity to print-related regularities and reading ability. Sensitivity to print-related regularities cannot develop without already having had some exposure to print; thus, it is unclear whether anything is gained by taking into account the sensitivity to regularities beyond simply assessing children’s reading ability.

In terms of treatment, there are three levels on which one could try to improve children’s reading ability. The first would be to train a domain-general statistical learning ability. To date, it is unclear whether this is possible or whether there would be any benefits even if a reliable training method is found. Second, one could increase children’s sensitivity to print-related statistical regularities by exposing them to more reading material. This would seem to be a sensible strategy in any case, as the number of books in a child’s house has been shown to be a predictor of reading ability, over and above genetic influences [93].

Third, one could explicitly teach children about regularities in their orthography. For example, if a child spells a word with an implausible letter sequence, such as *ckwick* for *quick*, one could explain that, even though *ck* is, indeed, a plausible spelling for the sound /k/, in English, words never start with *ck*. Similarly, one can teach a child to consider morphology in their spellings: if they spell, for example, *magician* as *majishun*, one can draw their attention to the relationship of this word to the word *magic* and other words containing the morpheme *ian*, such as *mathematician*. The focus is no longer on a cognitive statistical learning process but rather on explicitly teaching children print-related regularities. This raises the question of what is gained by invoking the concept of statistical learning in research on print-related regularities. From a theoretical perspective, it is relevant why a child has not been able to learn a particular regularity. From a practical perspective, however, it is likely that the treatment will be identical, regardless of the reasons why a child has not learned a given regularity: be it due to a statistical learning deficit, lack of exposure to reading materials, or a lack of explicit instructions relating to these regularities.

### 5.3. Further Theoretical Implications of Statistical Learning

As discussed in the Introduction, print consists mostly of orthographic regularities, and learning the statistical properties of these regularities is, by definition, necessary for reading acquisition. Thus, statistical learning is definitely critical for reading acquisition, but as this describes the very nature of an orthography, referring to reading as “statistical learning” arguably serves as a one-word explanation, which has a less well-defined meaning than the concept that it is proposed to describe, and shifts the focus away from the underlying cognitive mechanisms [94]. From a sceptic’s perspective, therefore, we propose to reserve the term “statistical learning” for the domain-general cognitive learning mechanism that is likely to play a role in many, if not all, learning tasks.

## 6. Conclusions

In the current review paper, we discuss issues related to the statistical learning deficit theory of dyslexia, some of which apply to distal deficit theories of dyslexia in general. First, we reviewed the evidence for a statistical learning deficit in dyslexia. While we find convincing evidence for a correlation, we argue that none of the existing studies establish causality: this is a concern for theories of dyslexia in general, which are often based on correlational or quasi-experimental evidence (i.e., differences in a given skill between participants with and without dyslexia). This issue can be addressed, at least in part, by future experimental and longitudinal work.

We then describe and model the noisy chain hypothesis, which could help in driving future studies. The hypothesis builds on the assumption that the strength of the relationship between a distal cause and reading ability or dyslexia is diluted with each intermediate linking skill. Our simulations show that this leads to a decrease in correlation between the most distal cause and reading ability with each additional intermediate link. Applied to (statistically) predicting whether a given child will show symptoms of dyslexia rather than predicting reading ability as a continuous variable, this should result in smaller group differences between participants with and without dyslexia for distal causes. The dilution of effect sizes is due to additional variables that influence each of the linking skills. This means that a larger number of participants is required to reach adequate statistical power compared to studies that test a proximal cause, which is closely linked to the end-state outcome of reading ability. For theories of distal causes of dyslexia, this means that much of the current research is likely to be underpowered. To circumvent issues with low statistical power, we propose (1) well-defined theories, which specify the number and nature of the intermediate links, (2) experimental research, where the influence of additional variables is minimised, and (3) studies focussing on adjacent links, which can be used to minimise measurement noise and to provide a proper effect size estimate to ultimately test the effect of the distal cause on reading ability or dyslexia.

## Figures and Tables

**Table 1 brainsci-11-01143-t001:** Results of simulations, showing the estimated correlation coefficient for different scenarios of relationship strength and number of intermediates.

Relationship Strength	Number of Intermediate Links	Correlation Coefficient Estimate	95% CI for Correlation Coefficient	Number of Participants for 80% Power
Low (0.2)	4	<0.001	−0.061, 0.062	185,187,455
Low (0.2)	3	0.002	−0.061, 0.064	1,843,597
Low (0.2)	2	0.009	−0.053, 0.070	92,274
Low (0.2)	1	0.049	−0.012, 0.111	3235
Low (0.2)	0	0.243	0.182, 0.301	130
Medium (0.5)	4	0.054	−0.007, 0.115	2657
Medium (0.5)	3	0.108	0.046, 0.169	675
Medium (0.5)	2	0.213	0.151, 0.272	170
Medium (0.5)	1	0.408	0.356, 0.458	44
Medium (0.5)	0	0.707	0.675, 0.737	12
High (0.8)	4	0.721	0.690, 0.750	11
High (0.8)	3	0.803	0.780, 0.824	9
High (0.8)	2	0.873	0.857, 0.887	7
High (0.8)	1	0.928	0.919, 0.937	6
High (0.8)	0	0.970	0.966, 0.974	5

**Table 2 brainsci-11-01143-t002:** Results of simulations, showing the estimated standardised effect size estimate for different scenarios of relationship strength and number of intermediates.

Relationship Strength	Number of Intermediate Links	Standardised Effect Size Estimate	95% CI for Effect Size Estimate	Total Number of Participants for 80% Power
Low (0.2)	4	−0.002	−0.134, 0.132	9,431,501
Low (0.2)	3	−0.008	−0.139, 0.123	482,294
Low (0.2)	2	−0.034	−0.165, 0.096	27,156
Low (0.2)	1	−0.144	−0.275, −0.013	1511
Low (0.2)	0	−0.589	−0.716, −0.464	92
Medium (0.5)	4	−0.429	−0.558, −0.300	172
Medium (0.5)	3	−0.607	−0.735, −0.480	87
Medium (0.5)	2	−0.859	−0.979, −0.738	45
Medium (0.5)	1	−1.215	−1.327, −1.101	23
Medium (0.5)	0	−1.717	−1.813, −1.621	13
High (0.8)	4	−2.087	−2.161, −2.014	10
High (0.8)	3	−2.151	−2.220, −2.083	9
High (0.8)	2	−2.217	−2.281, −2.154	9
High (0.8)	1	−2.285	−2.345, −2.227	8
High (0.8)	0	−2.356	−2.408, −2.305	8

## Data Availability

The code to the model can be found at https://osf.io/epu8m/, accessed on 27 June 2021.

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
