# Peer review of "Developmental Dyslexia, Reading Acquisition, and Statistical Learning: A Sceptic’s Guide"

_brainsci, 2021, doi:10.3390/brainsci11091143_

Round 1
Reviewer 1 Report
The authors present a review article which discusses the relationship between statistical learning and reading difficulties, presenting a summary of the literature, including problems along with the goal of describing the type of evidence needed to show a causal link between statistical learning efficiency and reading ability. The manuscript has a number of strengths, including the fact that it was clearly written and easy to follow. It also presents a brief summary of research in the field of SL in so much as it relates to learning to read and dyslexia. Despite this, I am not convinced that this review article offers much more than what researchers in this field already know.
Firstly, the authors start out by arguing that “focusing on sensitivity to orthography-specific regularities as a correlate of reading ability and dyslexia is more fruitful than focusing on a hypothetical domain-general statistical learning ability. (L14). However, I believe that the lead researchers in the field no longer argue for a domain general SL ability (see for example Frost et al., 2015. Domain generality versus modality specificity: The paradox of statistical learning). Furthermore, Siegelman, Bogaerts, & Frost (2017) presented a very persuasive argument as to why the field should move away from assessing group means and instead should focus on individual differences. But add to this the fact that current SL task are unreliable and do not have good psychometric properties (Arnon, 2019; 2020). Further, reviews such as Schmalz, Altoè & Mulatti (2017) provide explicit examples of poor practice in the field that could be improved. Thus, it seems to me that the issues raised in the present manuscript have been well covered. That said, I do acknowledge this is a review article, and the review presented is clear and precise.
Perhaps more importantly, one of the stated aims of the manuscript is (L39) to describe the type of evidence that would convince a sceptical researcher that there is a causal link between statistical learning efficiency and reading ability or dyslexia, as well as an evaluation of whether such evidence exists, to date. To my mind, the majority of the manuscript describes the state of the field, and highlights the many problems that exist, but describing what evidence would convince a sceptic doesn’t actually start until the section “Directions for future research” (L422). However, this very quickly turns to possible implications (L457), and what was presented in between was little more than “longitudinal data, rather than correlational data is needed to draw causal conclusions”. I found this underwhelming given that this is essentially what we aim to teach in undergraduate stats courses. The manuscript later returns to “the type of evidence that would convince” by raising issues such as publication bias and pre-registration. While these are important topics, they seemed really out of place to me. To clarify, the authors are quite right to suggest that these are sound practices that all competent research should observe. However, from the aims of the manuscript I was expecting some more domain specific recommendations rather than (sound) recommendations that apply to all fields.
In summary, I found the section which summaries the current state of the SL field with respect to learning to read and dyslexia to be clear and concise. Although all of the ideas presented have been covered elsewhere, such a revision may be useful for researchers who did not have the time to read the more extensive articles which have discussed these issues. However, the section of the manuscript dealing with the type of evidence needed to show a causal link between statistical learning efficiency and both reading ability and dyslexia was disappointing. In my opinion, although the issues presented were valid, there was little “domain specific” information and instead what was presented were a collection of good practices that apply to all fields. Accordingly, I think the manuscript fails to meet its stated objectives.
Minor points
L46 “If statistical learning were to play no role in reading at all”.
Even though it should be obvious to most readers, I think it would be desirable to distinguish between the act of learning to read (in which SL may have a role), and the act of “reading right now”, in which it is highly unlikely that SL has a role. So, the authors may want to consider changing this (and similar) phrase to “If statistical learning were to play no role at all in in learning to read”. It is not a major point I certainly am not going to insist that this change be made.
L72. “Third, statistical learning allows for generalisation: the ability to apply the learned knowledge to instances where it occurs in a different context. Thus, a child who has learned the pronunciation of the words “cat” and “let” will be able to apply the knowledge of the letter-sound correspondences to pronounce the unfamiliar pseudoword lat”.
I do not believe this to be a good example of the application of statistical learning. Wouldn’t reading aloud the word “lat” be more an example of where the rote-learned high-frequency letter-sound correspondences. I’m not criticising the point that SL allows for generalisation. I just think the specific example chosen to demonstrate the point is not a particularly good example. A better example would be something like the fact that the vowel in “PET” is usually spelled with “E”, but is usually spelled with “EA” if the following consonant is “D” (HEAD, SPREAD, DREAD, etc). This rule is not taught, but learned from exposure. Actually, I just realised my example is a spelling example, rather than a reading example. But it serves to demonstrate my point about when SL would come into play and I’m sure the authors will be able to come up with a reading example. Or perhaps just let the examples in the subsequent paragraph carry the explanatory burden.
L96. “The quasiregular relationship between orthography and semantics is less obvious”.
The authors may want to consider mentioning in this paragraph something from the body of work from Arciuli and colleagues who have shown that orthographic patterns are statistically linked to a word’s grammatical category and stress assignment (see for example Arciuli & Cupples, 2007; Ševa, Monaghan, & Arciuli, 2009). Other authors have published more recent work (for example, stress assignment; Ktori et al., 2018). My general point is only minor; that statistical regularities provide some other clues to pronunciation and semantics beyond those outlined in this paragraph.
490. “For the latter two treatments, the focus is no longer on a cognitive statistical learning process, but rather on teaching children print-related regularities. This raises the question of what is gained by invoking the concept of statistical learning in research on print-related regularities.”
Surely the point is that if a causal link was found between SL abilities and dyslexia, if one identified a child with impaired SL, it would be reasonable to assume that they may have difficulties in learning some important imbedded regularities in their language. This in turn could lead to difficulties in learning to read in typical learning settings. Hence, providing these children with an intervention which involved rote learning to overcome what a typically developing child would implicitly learn would seem to me to be a reasonable strategy. Thus, it does seem to me that there is something to be gained by studying the link between SL and print related regularities, that of identifying children at risk of not being able to learn in typical learning settings. This does not imply that all children with dyslexia will have an SL impairment, simply that some children who struggle to read (those with an SL impairment) may benefit from more explicit instruction.
References
Arciuli, J., & Cupples, L. (2007). Would you rather ‘embert a cudsert’ or ‘cudsert an embert’? How spelling patterns at the beginning of English disyllables can cue grammatical category. In A. C. Schalley & D. Khlentzos (Eds.), Mental states: Language and cognitive structure (Vol. 2, pp. 213-237). Amsterdam: John Benjamins.
Arnon, I. (2019). Statistical Learning, Implicit Learning, and First Language Acquisition: A Critical Evaluation of Two Developmental Predictions. Topics in Cognitive Science, 11(3), 504-519. doi: 10.1111/tops.12428
Arnon, I. (2020). Do current statistical learning tasks capture stable individual differences in children? An investigation of task reliability across modality. Behavior Research Methods, 52(1), 68-81. doi: 10.3758/s13428-019-01205-5
Frost, R., Armstrong, B. C., Siegelman, N., & Christiansen, M. H. (2015). Domain generality versus modality specificity: The paradox of statistical learning. Trends in Cognitive Sciences, 19(3), 117-125. doi: 10.1016/j.tics.2014.12.010
Ktori, M., Mousikou, P., & Rastle, K. (2018). Cues to stress assignment in reading aloud. Journal of Experimental Psychology: General, 147(1), 36.
Schmalz, X., Altoè, G., & Mulatti, C. (2017). Statistical learning and dyslexia: A systematic review. Annals of Dyslexia, 67(2), 147-162. doi: 10.1007/s11881-016-0136-0
Ševa, N., Monaghan, P., & Arciuli, J. (2009). Stressing what is important: Orthographic cues and lexical stress assignment. Journal of Neurolinguistics, 22(3), 237-249.
Siegelman, N., Bogaerts, L., & Frost, R. (2017). Measuring individual differences in statistical learning: Current pitfalls and possible solutions. Behavior Research Methods, 49, 418-432. doi: 10.3758/s13428-016-0719-z
Author Response
We thank R1 for their thoughtful comments! Below, we have copied each point and our response:
Point 1: The authors present a review article which discusses the relationship between statistical learning and reading difficulties, presenting a summary of the literature, including problems along with the goal of describing the type of evidence needed to show a causal link between statistical learning efficiency and reading ability. The manuscript has a number of strengths, including the fact that it was clearly written and easy to follow. It also presents a brief summary of research in the field of SL in so much as it relates to learning to read and dyslexia. Despite this, I am not convinced that this review article offers much more than what researchers in this field already know.
Firstly, the authors start out by arguing that “focusing on sensitivity to orthography-specific regularities as a correlate of reading ability and dyslexia is more fruitful than focusing on a hypothetical domain-general statistical learning ability. (L14). However, I believe that the lead researchers in the field no longer argue for a domain general SL ability (see for example Frost et al., 2015. Domain generality versus modality specificity: The paradox of statistical learning). Furthermore, Siegelman, Bogaerts, & Frost (2017) presented a very persuasive argument as to why the field should move away from assessing group means and instead should focus on individual differences. But add to this the fact that current SL task are unreliable and do not have good psychometric properties (Arnon, 2019; 2020). Further, reviews such as Schmalz, Altoè & Mulatti (2017) provide explicit examples of poor practice in the field that could be improved. Thus, it seems to me that the issues raised in the present manuscript have been well covered. That said, I do acknowledge this is a review article, and the review presented is clear and precise.
Perhaps more importantly, one of the stated aims of the manuscript is (L39) to describe the type of evidence that would convince a sceptical researcher that there is a causal link between statistical learning efficiency and reading ability or dyslexia, as well as an evaluation of whether such evidence exists, to date. To my mind, the majority of the manuscript describes the state of the field, and highlights the many problems that exist, but describing what evidence would convince a sceptic doesn’t actually start until the section “Directions for future research” (L422). However, this very quickly turns to possible implications (L457), and what was presented in between was little more than “longitudinal data, rather than correlational data is needed to draw causal conclusions”. I found this underwhelming given that this is essentially what we aim to teach in undergraduate stats courses. The manuscript later returns to “the type of evidence that would convince” by raising issues such as publication bias and pre-registration. While these are important topics, they seemed really out of place to me. To clarify, the authors are quite right to suggest that these are sound practices that all competent research should observe. However, from the aims of the manuscript I was expecting some more domain specific recommendations rather than (sound) recommendations that apply to all fields.
In summary, I found the section which summaries the current state of the SL field with respect to learning to read and dyslexia to be clear and concise. Although all of the ideas presented have been covered elsewhere, such a revision may be useful for researchers who did not have the time to read the more extensive articles which have discussed these issues. However, the section of the manuscript dealing with the type of evidence needed to show a causal link between statistical learning efficiency and both reading ability and dyslexia was disappointing. In my opinion, although the issues presented were valid, there was little “domain specific” information and instead what was presented were a collection of good practices that apply to all fields. Accordingly, I think the manuscript fails to meet its stated objectives.
Response: We have made several major changes to address these points. First, to ensure the match between our stated aims and the paper, we have rewritten the abstract and introduction. The statement of aims in the introduction section now reads:
“We examine the statistical learning deficit theory as a case study of a distal deficit hypothesis. First, we critically evaluate the evidence for a statistical learning deficit as a cause of dyslexia. Then, we turn to the more general question of what issues need to be resolved that are common to distal cause theories. Overall, this will provide directions for future research aiming to provide a comprehensive theory of the causes of dyslexia.” (now p. 4).
We have also added a new section to the manuscript, which we hope will provide some novel material, beyond a summary of the existing literature. We discuss and model the Noisy Chain Hypothesis (starting on p. 18), which applied not only to the statistical learning theory of dyslexia, but also to all other distal theories of developmental dyslexia.
Point 2: L46 “If statistical learning were to play no role in reading at all”.
Even though it should be obvious to most readers, I think it would be desirable to distinguish between the act of learning to read (in which SL may have a role), and the act of “reading right now”, in which it is highly unlikely that SL has a role. So, the authors may want to consider changing this (and similar) phrase to “If statistical learning were to play no role at all in in learning to read”. It is not a major point I certainly am not going to insist that this change be made.
Response: We have rephrased this sentence in accordance with R1’s suggestion (p. 4)
Point 3: L72. “Third, statistical learning allows for generalisation: the ability to apply the learned knowledge to instances where it occurs in a different context. Thus, a child who has learned the pronunciation of the words “cat” and “let” will be able to apply the knowledge of the letter-sound correspondences to pronounce the unfamiliar pseudoword lat”.
I do not believe this to be a good example of the application of statistical learning. Wouldn’t reading aloud the word “lat” be more an example of where the rote-learned high-frequency letter-sound correspondences. I’m not criticising the point that SL allows for generalisation. I just think the specific example chosen to demonstrate the point is not a particularly good example. A better example would be something like the fact that the vowel in “PET” is usually spelled with “E”, but is usually spelled with “EA” if the following consonant is “D” (HEAD, SPREAD, DREAD, etc). This rule is not taught, but learned from exposure. Actually, I just realised my example is a spelling example, rather than a reading example. But it serves to demonstrate my point about when SL would come into play and I’m sure the authors will be able to come up with a reading example. Or perhaps just let the examples in the subsequent paragraph carry the explanatory burden.
Response: We agree that the “lat”-example might not fully reflect a statistical learning mechanism, because children might use explicitly taught letter-sound knowledge for straight-forward pseudoword. We have changed the example to a similar one as R1 suggests, but for the print-speech direction: “Thus, a child who has learned the pronunciations of words like “wasp” and “swan” will read the pseudoword wamp as /wÉ”mp/ rather than /wæmp/, by generalising the context-sensitive grapheme phoneme correspondence [w]a ïƒ /É”/ to this new context.” (p. 5)
Point 4: L96. “The quasiregular relationship between orthography and semantics is less obvious”.
The authors may want to consider mentioning in this paragraph something from the body of work from Arciuli and colleagues who have shown that orthographic patterns are statistically linked to a word’s grammatical category and stress assignment (see for example Arciuli & Cupples, 2007; Ševa, Monaghan, & Arciuli, 2009). Other authors have published more recent work (for example, stress assignment; Ktori et al., 2018). My general point is only minor; that statistical regularities provide some other clues to pronunciation and semantics beyond those outlined in this paragraph.
Response: We have added the references (p. 6). It occurred to us that stress assignment is also relevant to the section on orthography-phonology regularities, so we have added a discussion of these regularities, as well as the more general point that the nature of regularities varies across orthographies, but that we assume that all orthographies contain regularities on some level (p. 14).
Point 5: 490. “For the latter two treatments, the focus is no longer on a cognitive statistical learning process, but rather on teaching children print-related regularities. This raises the question of what is gained by invoking the concept of statistical learning in research on print-related regularities.”
Surely the point is that if a causal link was found between SL abilities and dyslexia, if one identified a child with impaired SL, it would be reasonable to assume that they may have difficulties in learning some important imbedded regularities in their language. This in turn could lead to difficulties in learning to read in typical learning settings. Hence, providing these children with an intervention which involved rote learning to overcome what a typically developing child would implicitly learn would seem to me to be a reasonable strategy. Thus, it does seem to me that there is something to be gained by studying the link between SL and print related regularities, that of identifying children at risk of not being able to learn in typical learning settings. This does not imply that all children with dyslexia will have an SL impairment, simply that some children who struggle to read (those with an SL impairment) may benefit from more explicit instruction.
Response: We thank R1 for raising this interesting point! We have added a brief discussion about the theoretical versus practical benefits of knowing whether a lack of knowledge of a particular regularity is a result of a statistical learning deficit (p. 27).
Reviewer 2 Report
This is a well written and interesting ppaper. I am a sceptic too, but for a different reason, which, if discussed here, would make the paper even more interesting and anyway ought at least to be referred to. It could be, and often is, argued that most kinds of learning are anyway statistical in that a message has to be repeated many times to be learnt. But the key idea of those who apply the term to children learning to read is that they have to learn to extract abstract metaknowledge about relations between symbols and sounds that contributes to fluent reading and comprehension, and that this statistical process is somehow independent, generalises to other systems and seperable from the basic sensory and motor operations involved, rather like Chomsky's mythical 'encapsulated linguistic processor'. Much more likely is that the stats. emerge from brain's basic learning neurones' firing and wiring together, not as a seperate process, and any hint that dyslexics are worse at it, merely emerges from their deficient basic processing.
Author Response
Point 1: This is a well written and interesting paper. I am a sceptic too, but for a different reason, which, if discussed here, would make the paper even more interesting and anyway ought at least to be referred to. It could be, and often is, argued that most kinds of learning are anyway statistical in that a message has to be repeated many times to be learnt. But the key idea of those who apply the term to children learning to read is that they have to learn to extract abstract metaknowledge about relations between symbols and sounds that contributes to fluent reading and comprehension, and that this statistical process is somehow independent, generalises to other systems and separable from the basic sensory and motor operations involved, rather like Chomsky's mythical 'encapsulated linguistic processor'. Much more likely is that the stats. emerge from brain's basic learning neurones' firing and wiring together, not as a separate process, and any hint that dyslexics are worse at it, merely emerges from their deficient basic processing.
Response: We thank R2 for the overall positive review. We have added a discussion of this point (p. 8)